# Picking up the pieces: separately evaluating supernet training and architecture selection

## ABSTRACT

Differentiable Neural Architecture Search (NAS) has emerged as a simple and efficient method for the automated design of neural networks. Recent research has demonstrated improvements on various aspects on the original algorithm (DARTS), but comparative evaluation of these advances remains costly and difficult. We frame supernet NAS as a two-stage search, decoupling the training of the supernet from the extraction of a final design from the supernet. We propose a set of metrics which utilize benchmark data sets to evaluate each stage of the search process independently. We demonstrate two metrics measuring separately the quality of the supernet's shared weights and the quality of the learned sampling distribution, as well as corresponding statistics approximating the reliance of the second stage search on these components of the supernet. These metrics facilitate both more robust evaluation of NAS algorithms and provide practical method for designing complete NAS algorithms from separate supernet training and architecture selection techniques.

## 1 INTRODUCTION

Neural architecture search (NAS) describes the problem of automatically selecting an effective neural network architecture for given data. In their survey of work on NAS, Elsken et. al. establish three necessary components to achieve this goal: a search space to define the set possible architectures, a search strategy to traverse over potential solutions, and a performance estimation strategy to rate the quality of potential solutions considered by the search strategy (Elsken et al., 2019).

In recent years a variety of search spaces and benchmark problems have been proposed in NAS (Elsken et al., 2019). Exploring the space of search strategies has similarly generated vast interest, with approaches spanning evolutionary algorithms (Real et al., 2018), Bayesian optimization (Zela et al., 2018), reinforcement learning (Pham et al., 2018), and, most recently, direct gradient descent (Liu et al., 2019). Various methods for performance estimation have also been proposed, including weight inheritance (Real et al., 2017), predictive models (Liu et al., 2018), shortened training (Zoph et al., 2018), and weight-sharing (Pham et al., 2018).

However, the search space, search strategy, and performance estimation components of a NAS algorithm do not operate in a vacuum, and the interactions and relationships between them may dramatically affect the behavior and performance of the overall algorithm. For example, performance estimates that extrapolate from previously seen models are biased by the search strategy's method for traversing the search space and differentially sampling different models. Here, we suggest that the overwhelmingly trend in the current literature is to focus a study on innovations within a single one of these aspects (e.g. shared-weight model training for performance estimation) without a broader and systematic consideration, analysis, or even toolset for exploring the effects that this innovation may have on other components of the NAS algorithm (e.g. search strategy for selecting architecture topologies). While we do claim to provide a perfect solution for the analysis an algorithmic innovation's impact on each of the different components, here we provide one example of a systematic strategy and open-source toolset for performing such an investigation – and demonstrate a specific example of how such a perspective modifies the way that we interpret the theoretical impact of an important recent algorithmic and innovation.

This paper focuses specifically on the class of algorithms which combine a weight-sharing performance estimation strategy and a gradient-based search strategy into the training of a single model.

This approach trains a neural network representing a superposition of every architecture in the search space instead of training individual architectures and obtains performance estimates from this shared-weight model. This technique, which we will refer to as the supernet or supergraph method, was originally demonstrated by Pham et al. (2018), however the gradient-based training method contributed by DARTS (Liu et al., 2019) eliminated the need to train a second reinforcement learning model to select architectures, leading to the development of the series of NAS algorithms combining performance estimating and search in a single model considered here. Note that following Bender et al. (2018), the term "One-shot NAS" has been used to refer to supernet methods as they train a single model rather then iteratively training candidate architectures. However, as we will discuss iterative search algorithms implemented on top the supernet, we adopt the supernet terminology used by Wang et al. (2021). Differentiable supernet NAS algorithms have gained popularity due to their speed attained through weight-sharing – especially relative to employing the "gold standard" performance measure of training a neural network from multiple different random initializations and measuring the accuracy on a held-out set of validation data that often requires thousands of GPU hours for a single run (Zoph et al., 2018). They also benefit from and their ease of implementation within popular deep learning frameworks as a result of their reliance on gradient based optimization.

Importantly for this paper's perspective, these algorithms establish a special relationship between the search strategy and the performance estimation strategy – collapsing both components to the training of a single neural network. This presents a particular example of the cross-component interaction phenomenon described above during the supernet training, where the shared weights are trained to maximize the performance of the previously explored architectures, and are then employed to estimate the performance of new potential architectures. The feedback loop between these two processes thus creates an opportunity to bias search towards familiar solutions, possibly at the expense of undervaluing architectures which have high train-from-scratch performance, but may not perform well with the weights learned on previously explored architectures. This interaction is seen again post-supernet training, when the shared-weight model (itself embodying the chosen performance estimation strategy) is often critical in the process to select the single best architecture from the search space. With both architecture performance estimation and search strategy represented by the training of a single sueprnet model, it is perhaps not surprising that the distinct contributions of these two are not easily measured or conceptualized. We might further expect the degree to which this may be an issue to vary with the convexity of the search space, and the search strategy used to traverse that landscape.

In order to facilitate a greater understanding of, and more robust comparisons between, differentiable supernet NAS algorithms, here we present two statistics designed to separately evaluate (1) the architecture search and (2) the performance estimation components of the supernet training process. We also demonstrate the value of separately evaluating performance estimation and search during supernet training by presenting a two-stage framing of differentiable stochastic NAS, delinking the process of training the supernet from the process of selecting an architecture using the supernet. Using our statistics of the Stage-1 supernet training, we present a method for approximating the reliance of the Stage-2 architecture selection methods on the performance estimation capabilities of the supernet versus the discovery of quality architectures through supernet training. We offer empirical results demonstrating the capabilities of using the presented statistics to design NAS algorithms through the composition of separate supernet training and architecture selection methods, as well as highlighting a case-study demonstrating the importance of considering the interactions of Stage-1 supernet training and Stage-2 architecture selection on the analysis of a recent algorithmic innovation.

## 2 RELATED WORK

### 2.1 DIFFERENTIABLE SUPERNET NAS

DARTS was the first published differentiable NAS algorithm, utilizing a "continuous relaxation" method to parametrize the set of possible architectures, by placing trainable weights on each possible operation (Liu et al., 2019). The final model is selected from these trained architecture weights by simply keeping only the operations with the largest weight. This method is general enough to support the discovery of a broad range of neural network topologies through the use of a "zero" operation, representing the lack of an edge between nodes. Numerous works since the publication

of DARTS have attempted to demonstrate simple improvements on DARTS to stabilize training, avoid the hard prune at the end of training, or improve scalability. For example, PC-DARTS (Xu et al., 2020) has demonstrated partial channel training, allowing the use of larger supernet model. This work addresses the final hard prune by removing the magnitude based prune and substituting a perturbation-based pruning algorithm. While Chu et al. (2021) stabilizes the training process using an auxiliary skip connection, in contrast to prior work which relied on the Hessian eigenvalue regularizer.

In motivating SNAS, Xie et al. comment on the performance estimation aspect of DARTS, demonstrating the disparity in the validation accuracy obtained from the shared weight model of DARTS using the whole shared-weight model and the pruned final architecture (Xie et al., 2019). Stochastic differentiable NAS algorithms, like SNAS, offer one answer to this question of performance estimation in differentiable NAS through the use of sampling. These methods preserve the gradient-based approach of DARTS, while proposing a specific (and increasingly discrete) architecture at each step.

Motivating the performances estimation abilities of shared weight models more generally, Bender et al. (2018) implement a path-based form of dropout, randomly masking portions of the shared weight model during training and demonstrated a strong correlation between performance estimates from shortened training and the shared weight model, suggesting that the two methods of performance estimation are comparable.

## 2.2 ZERO-SHOT NAS

Recently, methods have been proposed to select an architecture without training even a single architecture model. Much of these methods are based on proxies, like EcoNAS (Zhou et al., 2020), while TE-NAS (Chen et al., 2021a) recently demonstrated a zero-shot search based on measures emerging from deep learning theory. Abdelfattah et al. (2021) demonstrate a range of measures which have been used to compute saliency in other deep learning problems as possible measures to be useful in NAS. In this work we modify the adapted synflow measure demonstrate by Abdelfattah et al. (2021) as well their implementation Jacobian covariance measure developed by Mellor et al. (2021) to function as architecture selection techniques on a supernet.

## 2.3 SUPERNET NAS EVALUATION

Research on the evaluation of NAS methods has developed as a result of difficulties in comparing the performance NAS algorithms demonstrated in different architecture search spaces, as the selection of search space can have a greater effect on the final performance than the selection of NAS algorithm, which can be made clear through performing a random search (Li & Talwalkar, 2019).

Yu et al. (2020) initiated the critical inquiry into the use of weight-sharing for performance estimation by evaluating popular weight-sharing NAS algorithms in a reduced search space of 32 architectures. They demonstrated a lack of correlation in ranking between the performance estimates obtained from the shared-weight model and the trained from scratch test accuracy for and an improvement in performance of the final model achieved by not using weight-sharing. Further research evaluating the rankings of small samples (Yang et al., 2020) or small search spaces (Zhang et al., 2020) showed high variance in rankings across random seeds after training the shared weight model through random sampling. Like these prior works, we also suggest the the accuracy of the top model alone can be a misleading evaluation technique for NAS algorithms. However, our basis for this assertion is not the design of the search space, but that the reported accuracy value may be due to the unintended interactions between algorithmic innovation presented and other components of the NAS algorithm (e.g. performance estimation or search strategy), and not necessarily well represent the merits of the proposed methodology itself. This motivates our proposal here for additional metrics which evaluate separately the performance estimation or architecture sampling aspects of a given algorithm.

This work is most aligned with direction of NAS evaluation research demonstrated by Zela et al. (2020). In this work they propose the dominant eigenvalue of the Hessian of the validation loss w.r.t. the architecture parameters as a statistic which helps to explain the failure modes of DARTS. Specifically, they associate a large eigenvalue with a significant drop in performance resulting from pruning. We are also attempt to establish statistics which indicate the potential outcome of archi-

tecture selection for a given trained supernet, however we are not focused on the pruning-based architecture selection used by DARTS. Instead of formulating our statistic to account for the failure modes of a specific NAS algorithm, we propose statistics based on the broad theory of the NAS process.

## 3 CASE STUDY: TWO MODIFICATIONS TO DARTS

To motivate the importance of separating the evaluation of different components of a NAS algorithm proposed below, here we present a experimental case study. Two recent papers demonstrated non-overlapping modifications to DARTS named DARTS- (Chu et al., 2021) and DARTS-PT (Wang et al., 2021). Each paper aimed to tackle the problem of unstable architecture weights in DARTS. DARTS- addressed this issue by adding an auxiliary skip connection to each mixed operation in DARTS, aiming to stabilize the training of architecture weights. DARTS-PT instead dismissed the architecture weights entirely, using a perturbation based method of selecting operations after training the supernet. The authors conclude that the perturbation technique may work better if DARTS is trained without architecture weights at all as they were able to demonstrate a smaller error using their method on a supernet trained with fixed architecture weights than on a supernet trained by DARTS. This result would suggest that the core innovation of DARTS, the continuous relaxation technique for approximating gradient updates to the architecture of the model, may not be of significant use. However, the generality of these conclusions are unclear from DARTS-PT alone.

| Search Space | DARTS | | | DARTS- | | |
|---|---|---|---|---|---|---|
| | Max $\alpha$ | Perturb | Perturb w/ fix $\alpha$ | Max $\alpha$ | Perturb | Perturb w/ fix $\alpha$ |
| NAS-Bench-201 | 45.7 | 11.89 | **6.20** | 6.75 | *6.56* | 7.22 |
| S1 | 3.84 | 3.5 | *2.86* | **2.62** | 2.87 | 2.76 |
| S2 | 4.85 | 2.79 | *2.59* | 2.65 | **2.47** | 2.80 |
| S3 | 3.34 | **2.49** | 2.52 | 2.58 | *2.53* | 2.54 |
| S4 | 7.2 | 2.64 | **2.58** | 3.44 | 2.67 | *2.61* |

Table 1: A case study showing the importance of separately evaluating supernet training and archtecture selection. The DARTS-PT architecture selection method run on a supernet trained on standard DARTS *(left)* shows improved peformance when not training architecture weights. But when the same perturbation method is applied to a supernet trained via DARTS- to results from including auxiliary skip connection *(right)*, we see improvement when including architecture weights. This reversal to the "Stage-2" perturbation method of selecting architectures is due to that method's reliance on the ("Stage-1") training of the supernet. The top method for each search space is indicated in bold while the top architecture selection technique for the version of DARTS which did not attain the overall top result is italicized.

By implementing DARTS- within the provided codebase for DARTS-PT and replicating their tests, we observe a reversal of this result in Table 1. We have run trials on CIFAR10 of perturbation search applied to DARTS- in each of the search spaces where DARTS-PT with a fixed $\alpha$ was shown to outperform DART-PT with a trained alpha (though this was only the case for S3 on data sets other than CIFAR10). Once training of the architecture weights is stabilized via the auxiliary skip connection, training with architecture weights actually does provide an improvement to the perturbation based selection method. Perturbation applied to DARTS trained with a fixed $\alpha$ still appears to be an effective method, attaining 2 of the top 4 results it held prior to comparison with DARTS- and the perturbation method appears quite effective overall, attaining the top result for every search space except S1. However, these results do suggest that the striking result that the process of training architecture weights actually results in a lower final performance after perturbation-based selection will likely not generalize beyond the original formulation of DARTS. Although the perturbation method does not utilize the architecture weights, the DARTS- results indicate that they influence the training of the shared weights in a way which can assist the perturbation process if the architecture weights are being trained effectively.

We do not present this case study in order to undermine the assertions supporting DARTS-PT about the inadequacy of architecture weight pruning as an architecture selection method. In fact, this

assertion aligns closely with the work demonstrated in this paper about alternative architecture selection techniques for differentiable NAS. Rather, we present this case study in order to demonstrate the necessity of a common theoretical frame in order to produce general insights to the architecture selection process in supernet NAS. In this case, an insight into the limitations of the architecture weights, turned out to no longer hold across a minor modification of the supernet training method. This specificity of the insight would have been difficult to anticipate as supernet methods lack a common interpretation of the architecture weights or method to evaluate them, beyond the performance of their argmax. We propose a general method for evaluating the supernet training and architecture selection methods separately as a path towards more general insights and rigorous understanding of supernet-based NAS algorithms.

## 4    Two stage search

Much of the criticism of DARTS has focused specifically on the method by which the final architecture is extracted from the supernet. These issues with the "hard-prune" implemented in DARTS distract from evaluation of the process by which DARTS trains the supernet. We view this as an extreme example of a broader phenomenon by which the reliance on the accuracy of final selected architectures as the standard performance metric dilutes our ability to evaluate specific innovations to the shared-weight model training and/or final architecture selection components of NAS. To remedy the lack of feedback on these individual aspects of a NAS algorithm, we employ a two stage search framework for differentiable supernet-based NAS.

We consider the training of the supernet as a "Stage-1 search algorithm" on its own. Although these algorithms do search the architecture space, their output is not a single final architecture but some form of information about the relative quality of architectures. We consider two possible forms this information can take: a weighting/sampling distribution over the architecture space (exemplified by $\alpha$ in DARTS) and a trained supernet model which able to provide performance estimates of individual candidate architectures. Note that the Stage-1 search algorithm need only provide one of these, as One-Shot NAS does not utilize any learned distribution over the architecture space, and the default Stage-2 search method for DARTS makes no use of the supernet model's shared weights.

We then denote the architecture selection process inherent to all supernet-based NAS algorithms as "Stage-2 search", which takes in the supernet model and associated weighted distribution over the architecture space and selects the single top architecture for that search space. The purpose of this terminology choice is to draw attention to distinct components of existing NAS algorithms and investigate their relationship to each other. There are numerous ways that a final architecture can be obtained from any supernet training algorithm and, though some selection algorithms may be more suitable for particular training algorithms, no supernet training algorithm is necessarily or trivially tied to a particular architecture selection algorithm, or vice versa. We utilize the original names of the algorithms to denote Stage-1 search algorithms for shared-weight model training, such as DARTS and SNAS, while we name the Stage-2 search algorithms after the architecture selection criterion used.

Note that these two search stages are simply a decomposition of the supernet-based search, and do not also consider or propose additional steps in the NAS search pipeline, such as obtaining train-from-scratch accuracy over multiple trials or further finetuning of the best of these models – as noted by Li & Talwalkar (2019) as aspects of the original DARTS search method.

Through this framing of two-stage supernet-based NAS, we are able to unify several disparate strands of NAS research. As the default Stage-2 search method of DARTS is notoriously unstable, techniques such as that of Wang et al. (2021) have been developed to bypass it. At the same time, zero-cost NAS methods, such as those of Abdelfattah et al. (2021) have provided advanced statistics for the rapid evaluation of sampled architectures. Positioning both of these techniques as Stage-2 search algorithms allows us to broaden the application of the former beyond DARTS and integrate the latter with supernet-based NAS techniques.

We evaluate both search stages and their combination using NAS-Bench-201 Dong & Yang (2020), relying on the train-from-scratch test accuracy metrics provided for each architecture in the benchmark. To begin to explore the generality of these results, we provide results on the three provided benchmarks within the dataset (CIFAR-10, CIFAR-100, and ImageNet16-120). Cognizant of the rel-

atively small search space of this setup, we also provide results on CIFAR-10 on the larger DARTS search space.

## 4.1 EVALUATING STAGE-1 SEARCH

We consider the process of training a supernet to be Stage-1 of the architecture search, resulting in a relative ranking estimating the quality of architectures (or their subcomponents of individual edges/nodes/operations). How much this process actually constitutes a search of the architecture space depends entirely on the training method. If we were to substitute training a subset of the supernet for individual model training in an evolutionary NAS algorithm, we would obtain a training method where the supernet is trained via an explicit search of the architecture space. Along these lines, the Random Search with Parameter Sharing (RSPS) (Li & Talwalkar, 2019) training method can be viewed as a random search of the architecture space or, conversely a supernet training method which involves no search of the architecture space (as the information about the observed performance of sampled architectures is not used to inform the search process). The algorithms we consider here, differentiable supernet training methods, occupy a more ambiguous position. DARTS and its variants do not evaluate individual architectures in the search space during training, but do evaluate supersets of components of the search space. In NAS-Bench-201, and any other search space encoded as a set of single-choice categorical variables, the sampling-based differentiable NAS algorithms actually do iterate over architectures within the search space, though only those that impose a discrete constraint like GDAS are actually evaluating individual architectures. In general, these differentiable supernet training methods can be construed as implicitly searching over the architecture space, performing and retaining record of comparative evaluations within the search space, but not necessarily at the level of individual architectures.

We wish to, as separately as possible, evaluate how these algorithms train their architecture and shared weights. As NAS-Bench-201's search space allows a single operation at each edge, the architecture weights, after a softmax, correspond exactly to a categorical distribution over operations. From the probability of each operation in a model, we can take their product to find the sampling probability mass assigned to each architecture in the search space. We utilize the benchmark data set to compute the probability of sampling architectures of a given test accuracy, allowing us to evaluate the quality of the architecture weights by directly observing the sampling probability mass they have assigned to the highest performing models in the search space. Fig. 1 *(Right)* shows the cumulative distribution of sampling probability by test accuracy. Points above the baseline of uninformed random sampling (dashed line) demonstrate algorithms with a tendency to oversample high performing architectures, while curves below the baseline undersample high performing architectures.

In order to evaluate the quality of the shared weights learned by the supernet training algorithms, we utilize them to record a validation accuracy for every architecture in the search space. We then rank every architecture according to its approximated performance and compare this ranking to the ground truth ranking obtained using the benchmark test accuracy. For a given test accuracy level, the proportion of architectures with ground truth accuracy of at least that level that are also predicted to be in that ranking is recorded (e.g. if 20% of the architectures in the top decile of models according to their train-from-scratch test accuracy are predicted to be in the top decile via shared-weight validation accuracy for a given algorithm, then at the test error threshold separating the top decile of architectures from the bottom 90%, that algorithm would have a "top-architecture-identification" (i.e. recall) rate of 20% or 0.2). Fig. 1 *(Left)* shows this curve relative to a baseline of random performance-estimation validation rankings. Points above the baseline better estimate performance for higher-performing architectures, while points below the baseline curve are worse at estimating the performance of higher-performing architectures. In addition to the CIFAR-10 performance estimation and sampling probability curves Fig. 1 *(Top)*, the metrics on Imagenet Benchmark from NAS-Bench-201 in Fig. 1 *(Bottom)* show that SNAS and Dirichlet Neural Architecture Search (Dr-NAS) highly outperform the other algorithms at both estimating performance of top architectures and biasing the sampling distribution towards them. Interestingly, DARTS (which never samples discrete architecture during the training of the shared-weight model) performs very poorly at both performance estimation and oversampling of top architectures. With the DARTS- innovation, the supernet does improve its ability to oversample high-performing architectures, making the algorithm competitive with the worse performing sampling-based approach we investigated (GDAS).

Although only one Stage-2 method we evaluate actually directly relies on validating individual architectures using the supernet, by obtaining performance estimates for each architecture in the search space and then attempting to rank them, we provide a proxy measure the amount of information about the relative quality of individual architectures within the supernet, which should also play a role in determining any other statistic computed over individual architectures using the supernet.

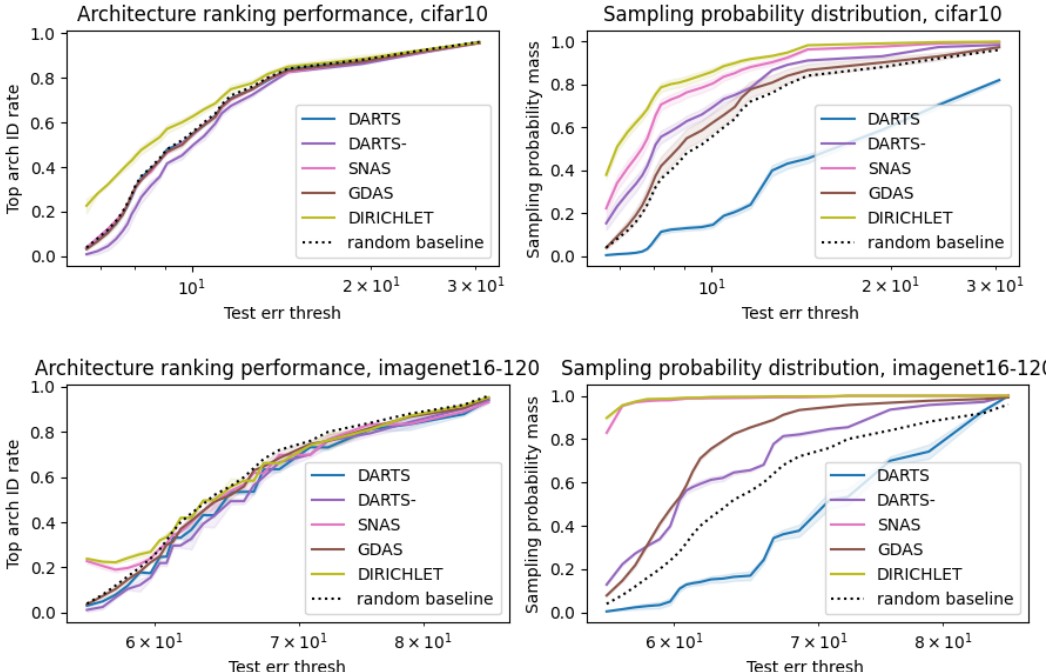

Figure 1: *(Left)* Top Arch ID rate of shared-weight based validation accuracy and benchmark test accuracy across test error threshold groups of the architecture space. *(Right)* Sampling probability mass across test error threshold groups of the probability space.

## 4.2 EVALUATING STAGE-2 SEARCH

In our Stage-2 metric, we seek to evaluate the ability of an algorithm to select the top discrete architectures from a given supernet. In order to evaluate the architecture selection process, we define a standardized input for the Stage-2 search algorithms. We wish to provide a sufficiently broad baseline such that it is suitable for evaluating any Stage-2 search algorithm which can be implemented on top of a supernet training algorithm. As we consider the supernet itself as the input to the architecture selection algorithms, our baseline must consist of a standardized set of supernets, trained in a manner such that evaluating architecture selection methods on them facilitates estimation of their efficacy across a range of Stage-1 search methods.

Unfortunately, there is no readily apparent method for training architecture weights that could serve as a suitable baseline. As the method by which they train the architecture weights is the defining characteristic for several of the Stage-1 search algorithms we examine and the simplest method to train the architecture weights is already one of the algorithms (DARTS) it is not feasible to design a baseline training method which involves training the architecture weights without biasing the baseline towards evaluating compatibility with a specific Stage-1 algorithm.

The core of our baseline Stage-2 evaluation technique is given by the algorithm "Random Search with Parameter Sharing" (RSPS) proposed by Li & Talwalkar (2019) as a baseline method for evaluating the contribution of weight-sharing in NAS. Additionally, sampling architectures uniformly at random and training the supernet by adjusting the weights for a single architecture at a time presents the simplest method for training a supernet.

In order to simulate the benefit of architecture weights without actually training architecture weights, we utilize the relationship between the architecture weights and architecture sampling. As we are using a NAS benchmark to support this evaluation method, we have access to the ground truth performance of every architecture in the search space and can design an arbitrarily good distribution over the architecture space. As such, we augment our RSPS training method by replacing the uniform sampling with a sampling distribution where the probability of sampling a model is inversely proportional to its performance (so the sampling probability linearly decreases over the ranking by test accuracy), formulating and idealized version of discrete sampling based supernet NAS. We also utilize this distribution for evaluation of Stage-2 search algorithms which sample architectures from the supernet.

In addition to the RSPS model evaluated with ($R_I$ and without $R_U$ the biased sampling distribution), we run each Stage-2 search algorithms on a randomly initialized model with and without the idealized sampler ($N_I$ and $N_U$ respectively). We compute statistics corresponding to the reliance of the Stage-2 algorithm on the architecture discovery and performance estimation via supernet shared weights by taking the difference in performance on these baseline models. The measure corresponding to reliance on the supernet having learned effective architecture parameters is computed as the difference between the performance of the biased sampling supernet and the uniform sampling supernet. The measure corresponding to reliance on obtaining information about the effectiveness of individual candidate architectures using the supernet's weights, is computed as the difference between performance on the uniform sampling supernet and the randomly initialized supernet.

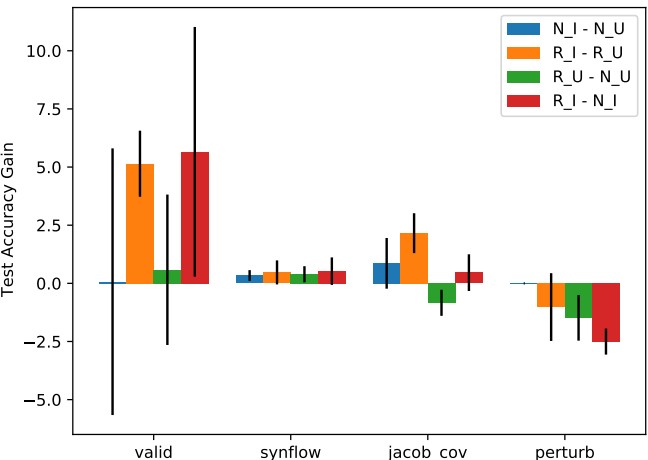

Figure 2: Comparison of Stage-2 search algorithm performance with given baseline models. "Initial" refers to untrained model, "Uniform" refers to a model trained by training a uniformly randomly sampled architecture at each batch, and "Biased" refers to a model trained by an architecture sampled from an arbitrarily good distribution at each batch.

We examine four techniques for Stage-2 search: searching based on the validation accuracy obtained from the shared weight model ("valid"); the top two measure reported in Abdelfattah et al. (2021), synflow and jacob_cov, and the perturbation based method demonstrated in Wang et al. (2021) ("perturb"). The first 3 methods are sampling based random searches: a sample of 100 architectures is drawn from the architecture distribution provided by the supernet and then the candidate architectures are ranked by either the validation accuracy, synflow, or jacobian covariance, outputting the highest ranked architecture. The perturbation based method instead operates directly on the supernet, iteratively masking out operations and computing validation accuracies to prune operations one by one until a final model is produced.

## 5 Putting it Together

The concept of separate Stage 1 and Stage 2 search algorithms hinges on the assumption that these algoritm are not closely coupled in practice: that a better result may be obtained by swapping the Stage 2 search algorithm that was originally paired with a given Stage 1 search algorithm in an appropriate context.

The results of evaluating all of the reviewed Stage 1 and Stage 2 search algorithms in combination seem to support his assumption.

|         | prune | valid | synflow | jac cov | perturb |
|---------|-------|-------|---------|---------|---------|
| darts     | $40.16 \pm 7.83$ | $\mathbf{9.07} \pm 1.65$ | $9.48 \pm 2.33$ | $10.03 \pm 0.20$ | $14.34 \pm 3.66$ |
| darts-    | $9.36 \pm 2.17$ | $11.90 \pm 0.65$ | $\mathbf{6.29} \pm 0.46$ | $8.16 \pm 2.03$ | $10.17 \pm 1.16$ |
| snas      | $\mathbf{5.92} \pm 0.25$ | $8.31 \pm 3.36$ | $6.36 \pm 0.10$ | $7.71 \pm 0.27$ | $7.24 \pm 1.54$ |
| gdas      | $12.55 \pm 8.54$ | $17.40 \pm 4.39$ | $\mathbf{7.03} \pm 0.20$ | $9.30 \pm 1.26$ | $8.25 \pm 1.46$ |
| dirichlet | $\mathbf{5.64} \pm 0.00$ | $6.35 \pm 0.08$ | $6.04 \pm 0.28$ | $7.35 \pm 0.75$ | $5.88 \pm 0.22$ |

Table 2: Combined results of all S1 and S2 search algorithms on CIFAR-10 in NAS-Bench-201

We also evaluate each Stage-1 and Stage-2 algorithm and their combinations on CIFAR100 and ImageNet16-120, with the result provided in the Appendix.

### 5.1 DARTS-space results

|         | prune | valid | synflow | jac cov | perturb |
|---------|-------|-------|---------|---------|---------|
| darts     | $3.27 \pm 0.44$ | $3.48 \pm 0.47$ | $3.66 \pm 0.59$ | $\mathbf{3.01} \pm 0.23$ | $3.25 \pm 0.10$ |
| darts-    | $\mathbf{2.66} \pm 0.15$ | $2.98 \pm 0.22$ | $3.22 \pm 0.43$ | $3.11 \pm 0.40$ | $2.72 \pm 0.25$ |
| snas      | $3.10 \pm 0.14$ | $3.19 \pm 0.31$ | $3.61 \pm 0.67$ | $3.13 \pm 0.35$ | $\mathbf{2.96} \pm 0.32$ |
| gdas      | $3.04 \pm 0.39$ | $4.01 \pm 0.62$ | $4.94 \pm 0.29$ | $3.17 \pm 0.23$ | $\mathbf{2.90} \pm 0.11$ |
| dirichlet | $\mathbf{2.65} \pm 0.06$ | $3.36 \pm 0.07$ | $3.67 \pm 0.66$ | $3.21 \pm 0.32$ | $2.78 \pm 0.05$ |

Table 3: Combined results of all S1 and S2 search algorithms on CIFAR-10 in DARTS-space

## 6 Conclusion

In order to facilitate integration of recent advances in NAS and well as more robust evaluation in NAS research, we have reframed existing differentiable supernet NAS algorithms as combinations of two separate search algorithms, the first being an implicit search via the process of training the supernet using architecture weights and the second being the process of selecting an architecture using the supernet. We propose two statistics approximating the quality of the weighting over the architecture space and the capacity of the supernet to provide information about the relative quality of individual candidate architectures, as well as two corresponding statistics approximating the reliance of architecture selection methods on each of these components of the supernet.

We measure the performance of each of 5 Stage-1 and 5 Stage-2 search methods in combination, noting that we did not find consistent combinations of algorithms optimal across different datasets, search spaces, and training conditions. We did find, however, that our statistics did elucidate properties of the Stage-1 search algorithms which achieve top performance across search spaces, like the capacity of SNAS or DirichletNAS to estimate performance, that distinguished them from less successful Stage-1 search algorithms.

## 7 Reproducibility

We trained each model on a single Tesla V100 GPU. We utilized the random seeds [1, 2, 3] in our case study and [10, 11, 12] in our main experiment. The source code utilized in the course of this experimentation is provided in full at: https://github.com/anon-submit-ml/picking-up-pieces

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

# A APPENDIX

## A.1 REVIEW OF SEARCH 2 METHODS IN PRIOR WORK

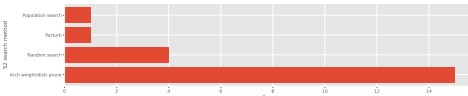

Figure 3: Stage-2 search methods of published NAS algorithms

We review a set of 20 published supernet NAS algorithms to determine the nature of the Stage-2 search method used. We find that existing techniques overwhelmingly utilize some form of architecture weight or distribution pruning to select a final architecture.

| Algorithm | S2 Search method | Notes | Reference |
|---|---|---|---|
| DARTS | Arch prune | Suggests multiple trials for final arch | 2019 |
| DARTS- | Arch prune | | 2021 |
| PC-DARTS | Arch prune | Combined prune on op, edge weights | 2020 |
| RSPS | Random search | | 2019 |
| SNAS | Arch prune | Prune ops, no method for edge pruning | 2019 |
| GDAS | Arch prune | | 2019a |
| DrNAS | Arch prune | | 2021b |
| ProxylessNAS | Arch prune | Sample during training | 2019 |
| ENAS | Random search | Uses trained sampler | 2018 |
| OSNAS | Random search | | 2018 |
| DARTS-PT | Perturbation | | 2021 |
| P-DARTS | Arch prune | Progressive prune, alter search space | 2019 |
| SDARTS | Arch prune | Arch selection not directly addresssed | 2021 |
| R-DARTS | Arch prune | Final arch selected from multiple trials | 2020 |
| SETN | Random search | Uses trained sampler | 2019b |
| BayesNAS | Arch prune | Entropy-based prune | 2019 |
| NASP | Arch prune | Proximal algorithm, applied during training | 2020 |
| SGAS | Arch prune | Progressive prune on multiple metrics | 2020 |
| NAOnet | Population search | Gradient based updates to population | 2018 |
| PARSEC | Arch prune | Sample during training | 2019 |
| DSNAS | Arch prune | Sample during training | 2020 |

Table 4: Detailed account of Stage-2 Search method review.

In the above table, note that when "sample during training" is included in the notes, this signifies that discrete architectures are sampled during the training process, while the final architecture selection method does not depend on sampling. This is technically not true for GDAS, as it samples a single operation on each edge in same search space utilized by DARTS which does not allow an operation on every edge, resulting in sampling a subset of the supernet that is not necessarily equivalent to any architecture within the search space. SNAS and DrNAS also utilize sampling during training, but use relaxed samples, not discrete architectures.

As clarified in the notes, there is significant variation among the architecture techniques labeled as "arch prune." Many methods utilize more complex parameterizations of the architecture space than DARTS, but largely still prune based on magnitude. BayesNAS instead utilizes the entropy of the architecture parameterization as a pruning criterion, while SGAS develops two different criteria incorporating magnitude, entropy, as well as the histogram intersection of the architecture parameterization over several updates. The technique used by NASP is a proximal algorithm, which in the general case would be incorrect to describe as pruning, however the proximal algorithm used in this case applies the constraint of the $L^0 - ball the resulting process is similar$.

## A.2 CIFAR-100 RESULTS

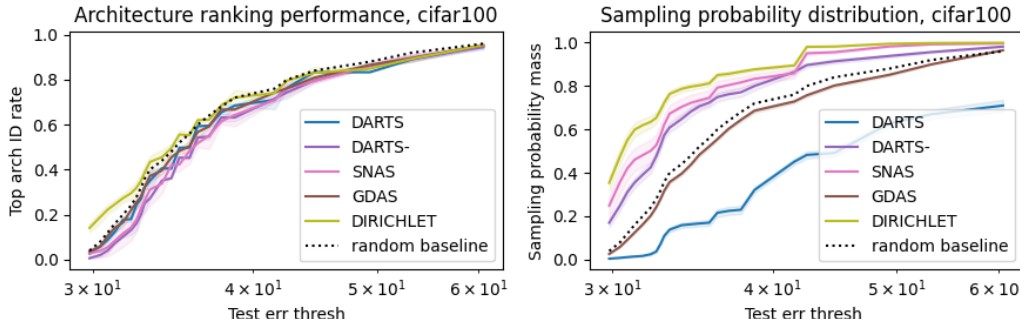

Figure 4: Left: Top arch ID rate across test regret threshold groups of the architecture space. Right: Sampling probability mass across test regret threshold groups of the probability space.

|  | prune | valid | synflow | jac cov | perturb |
|---|---|---|---|---|---|
| darts | $61.03 \pm 0.00$ | $36.41 \pm 3.19$ | $33.24 \pm 33.24$ | $36.44 \pm 1.15$ | $44.13 \pm 5.60$ |
| darts- | $32.46 \pm 4.93$ | $44.68 \pm 7.47$ | $28.51 \pm 28.51$ | $32.18 \pm 2.46$ | $35.11 \pm 3.55$ |
| snas | $28.37 \pm 1.33$ | $36.95 \pm 6.32$ | $29.08 \pm 29.08$ | $32.47 \pm 3.77$ | $33.65 \pm 3.35$ |
| gdas | $83.58 \pm 8.80$ | $52.97 \pm 7.97$ | $30.21 \pm 30.21$ | $36.20 \pm 2.74$ | $38.81 \pm 4.05$ |
| dirichlet | $28.62 \pm 1.57$ | $30.03 \pm 0.56$ | $27.48 \pm 27.48$ | $34.44 \pm 3.77$ | $29.29 \pm 2.43$ |

Table 5: Combined results of all S1 and S2 search algorithms on CIFAR-100 in NAS-Bench-201

## A.3 IMAGENET16-120 RESULTS

|  | prune | valid | synflow | jac cov | perturb |
|---|---|---|---|---|---|
| darts | $81.59 \pm 0.00$ | $79.21 \pm 5.36$ | $57.41 \pm 0.90$ | $71.52 \pm 4.47$ | $65.04 \pm 1.22$ |
| darts- | $77.03 \pm 4.70$ | $77.49 \pm 5.21$ | $54.68 \pm 1.44$ | $61.61 \pm 5.11$ | $64.65 \pm 3.03$ |
| snas | $53.66 \pm 0.00$ | $53.44 \pm 0.11$ | $56.93 \pm 2.31$ | $54.64 \pm 0.76$ | $56.20 \pm 2.97$ |
| gdas | $58.98 \pm 0.00$ | $69.26 \pm 4.79$ | $58.62 \pm 0.82$ | $63.39 \pm 4.54$ | $59.01 \pm 2.13$ |
| dirichlet | $53.66 \pm 0.00$ | $54.68 \pm 0.17$ | $53.66 \pm 0.00$ | $56.15 \pm 1.46$ | $56.04 \pm 3.07$ |

Table 6: Combined results of all S1 and S2 search algorithms on ImageNet16-120 in NAS-Bench-201

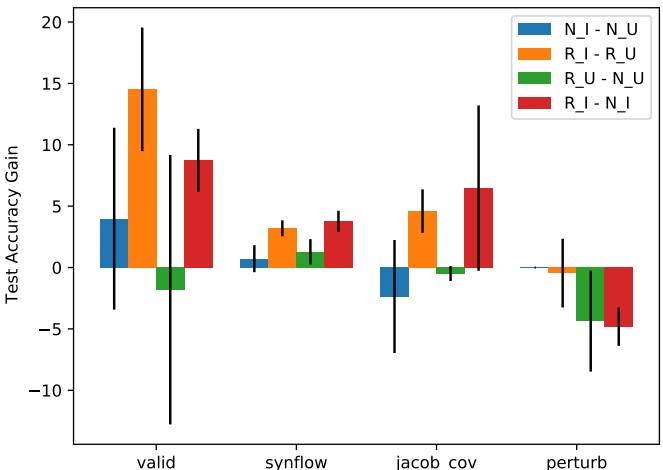

Figure 5: Comparison of Stage-2 search algorithm performance on CIFAR-100 with given baseline models. "Initial" refers to untrained model, "Uniform" refers to a model trained by training a uniformly randomly sampled architecture at each batch, and "Biased" refers to a model trained by an architecture sampled from an arbitrarily good distribution at each batch.

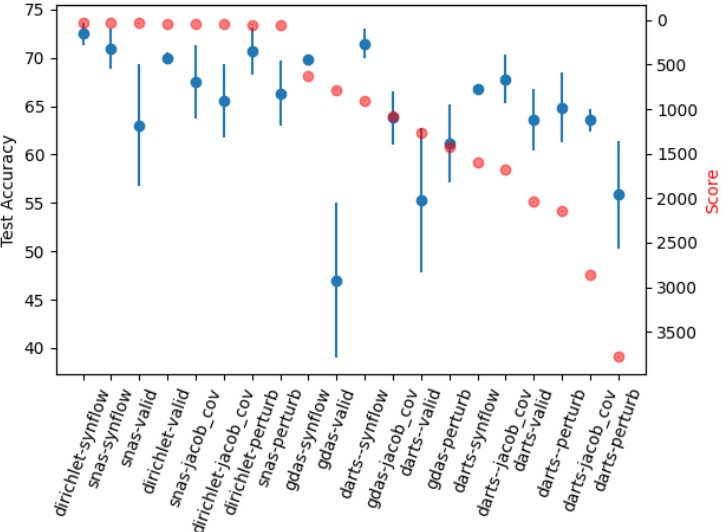

Figure 6: Performance of Stage-1 and Stage-2 search on CIFAR-100 algorithm combinations, ordered by score.

## A.4 PREDICTING COMBINED PERFORMANCE

Separate measures for evaluating Stage-1 and Stage-2 search facilitate estimation of the quality of combining Stage-1 and Stage-2 methods. To empirically test the utility of these measures, we combine the separate evaluation methods via a simple ranking technique to generate a unified ranking.

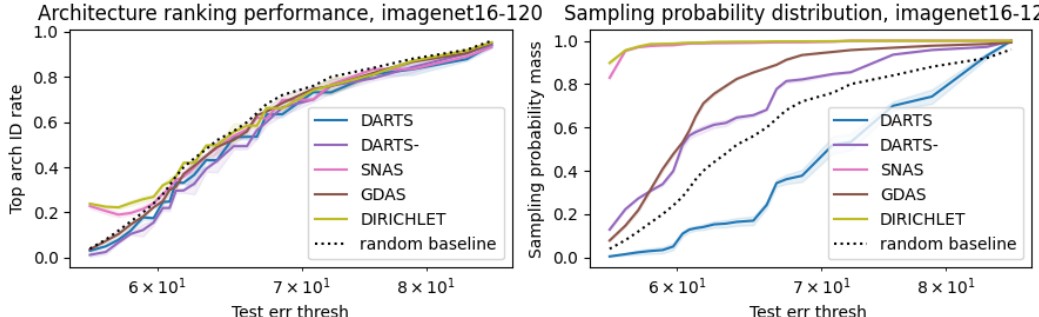

Figure 7: Left: Top arch ID rate across test regret threshold groups of the architecture space. Right: Sampling probability mass across test regret threshold groups of the probability space.

We compared this estimated ranking based on separate evaluation to the observed performance of each combination of search Stage-1 and search Stage-2 implemented on NAS-Bench-201.

We denote the ranking of Stage-1 search algorithms by sampling probability as $r_{\alpha 1}$ and the ranking by correlation of shared-weight validation accuracy with benchmark test accuracy as $r_{\beta 1}$. For Stage-2 models we compute the performance on the uniform baseline as $r_{\alpha 2}$ and the performance on the biased baseline as $r_{\beta 2}$. Then we compute a score for each combination of Stage-1 and Stage-2 algorithm as follows:

$$score = r_{\alpha 1}(|r_{\alpha 1} - r_{\alpha_2}| + 1) \times r_{\beta 1}(|r_{\beta 1} - r_{\beta 2}| + 1)$$

The scoring formula uses the difference between ranks to capture a mismatch in the capabilities of Stage-1 algorithms and those relied on by the Stage-2 search algorithms, adding 1 to avoid attributing undue weight to the Stage-1 and Stage-2 algorithms sharing a rank. We then scale this difference by the Stage-1 ranks as they are approximations of the supernet model's performance estimation and top architecture discovery capabilities, which should correlate with overall performance. As the Stage-2 ranks are computed using the difference in performance between pair of models rather than being a measure of model performance, we do not incorporate their magnitude. Given that NAS algorithms have been demonstrated which do not rely one of either a learned distribution over architectures or ever using the supernet to evaluate individual architectures, we use multiplication to combine the two separate measures so that a combination which succeeds at one of the two measures is scored above one which is mediocre at both.

After computing the scores, we then test the Stage-2 search algorithms on each Stage-1 algorithm, producing measurements of the benchmark test accuracy of the final architecture we are able to compare against the ranking implied by our scores, as shown in Fig. 7.

These final scores, with the accompanying ordering, suggest that these measures are a useful, but imperfect message for estimating the combined performance of a supernet training algorithm and architecture selection algorithm.

The purpose of this provided ranking is not necessarily to propose a specific method for ranking combinations of Stage-1 and Stage-2 algorithms but rather to provide an empirical demonstration of the usefulness of the proposed evaluation techniques and broader perspective of evaluating separately the performance estimation, architecture sampling, and architecture selection. These evaluation techniques can be utilized by researchers who develop new Stage-1 or Stage-2 algorithms to provide additional information about how their algorithm functions and how it might interact with other NAS techniques that is not provided when only the final test accuracy is reported.

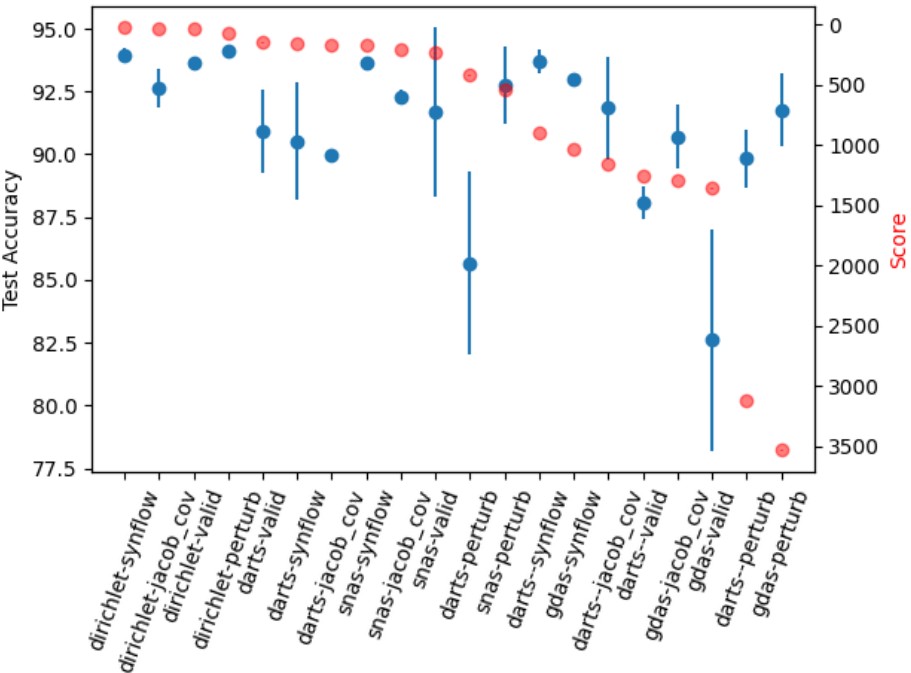

Figure 8: Performance of Stage-1 and Stage-2 search algorithm combinations, ordered by decreasing score, with benchmark test accuracy plotted in blue on the left axis and score plotted in red on the right axis.

