# OpenReview forum: "Picking up the pieces: separately evaluating supernet training and architecture selection"
_ICLR.cc/2022/Conference — ICLR 2022 Submitted_

### Official Review · Reviewer_dryA · 2021-10-26

**Correctness:** 3
**Technical Novelty And Significance:** 2
**Empirical Novelty And Significance:** 2
**Recommendation:** 3
**Confidence:** 4

**Main Review:**


-- Strength --
+ Clearly defining two stages of DARTS method is not novel in terms of research, however, this paper is written in a clear fashion and is good for new practitioner to NAS DARTS field to quickly understand the limit of DARTS based methods.
+ Study different combinations of DARTS based methods in terms of their super-net training and evaluation is kind of interesting, and the newly proposed score seems to be efficient in their preliminary experiment on NASBench-201.



-- Weakness --

- Overall the content is interesting but quite preliminary. NASBench-201 is a very small search space, (only has ~6k unique architectures). The findings on this small space may not generalize to realistic ones (e.g. DARTS space). In other words, I suggest the authors consider using recent NASBench-301 surrogate benchmarks to perform solid test to see the findings can or cannot generalize.

All figures except Figure 2 are quite large, and can be shrunk to a smaller size. This also indicates the paper seems to have not sufficient content to meet the ICLR standards. As an empirical evaluation paper, lacking enough juice for the community is a critical weakness.


- Questions about ranking correlation computation on DARTS-based supernet

As we know, the differentiable architecture search approach defines a super-net as weighted summation, e.g. if we have two choice branch, b1, and b2, and architecture parameter \alpha = [a1, a2], the output is usually equal to \sum ai * bi. In this case, the super-net training depends on \alpha and b1, b2. And in my experience, the a_i \in [0,1] during training, and often is quite close to 1/n (n = number of operations) especially during the warm-up phase.

However, the final architecture in NASBench-201 only contains 1 operation, i.e. ai \in {0, 1} as a discrete value. In this case, if we want to select b1, b2 separately to compute the ranking correlation, do the authors set a1 = 1 and a2=1 accordingly? Is this really meaningful? Since the super-net is never trained with a1=1 or a2=1? This detail greatly impact the credibility of Figure 2 result. And I did not find much detail about this implementation.

Speaking of which, such ranking correlation analysis of DARTS-based method is done in Yu et al. Landmark regularization: Ranking guided super-net training in NAS, CVPR2021. Please see appendix Fig 1 for more details. (https://openaccess.thecvf.com/content/CVPR2021/supplemental/Yu_Landmark_Regularization_Ranking_CVPR_2021_supplemental.pdf)


**Summary Of The Paper:**

This paper analyzed previous DARTS-based method and summarize these approaches into two stage, stage-1 that mainly trains the super-net, and stage-2 where the sampler try to select the best architecture out of the super-net. This paper proposes to evaluate the previous methods separately on NASBench-201.


**Summary Of The Review:**

see above

---

> ### Author Response · Authors · 2021-11-23
> **Response to Reviewer dryA**
>
> We thank the reviewer for helpful feedback and comments.  In response to the noted weaknesses we have:
>  - We have expanded the experiments to also include the ImageNet16-120 benchmark, as well as the larger DARTS-space search space on CIFAR-10.
> - We have reformatted and resize figures, and use the space to expand our results and analysis
> - We have replace the supernet correlation computation with a simpler and more intuitive method
> - We have expanded our literature review to include citations missing previously
> - You were correct in your observation that in validating the trained supernet with all possible candidate architectures we are using discrete architecture weights not used during training (except possibly for GDAS). The fairness of this is not necessarily given, however, as we are evaluating mostly supernet NAS algorithms which employ continuous relaxation, including relaxed sampling of the architecture space, which architectures the algorithm has "seen" during training is not a question with a straightforward answer. These sampling based algorithms approach, but do not reach the discrete encoding of architectures during training, which is likely why they alone seem to demonstrate success at performance estimation as measured this way

---

### Official Review · Reviewer_XbXs · 2021-10-26

**Correctness:** 4
**Technical Novelty And Significance:** 2
**Empirical Novelty And Significance:** 2
**Recommendation:** 3
**Confidence:** 5

**Main Review:**

Strengths:
- Developing better strategies to evaluate NAS algorithms is an interesting topic. Although it has received an increasing attention in the past couple of years, there remain open questions.
- The proposed separation of differentiable NAS methods into two stages is reasonable.

Weaknesses:

1) Clarity: Altogether, I found the paper difficult to follow; it would benefit from a thorough revision.
1.1) I did not quite understand what the paper was about (beyond differentiable NAS evaluation) from the introduction.

1.2) I struggled to follow the description of the evaluation process for Stage 1 and Stage 2 (Sections 4.1 and 4.2).

1.3) The motivation and description of the score in Section 5 remains unclear to me. For example, is $r_{\alpha_1}$ a vector (as would the term "ranking" suggest), or a single value? Note that $\beta_1$ was not defined in Section 4 (only $\alpha_1$, $\alpha_2$ and $\beta_2$).

2) Discussion of the results:
2.1) In Section 4.1, the curves in Figure 2 (left) are briefly mentioned, without any discussion of what these curves mean. The plot in Figure 2 right is not even mentioned. I strongly recommend the authors to comment on these results.

2.2) Similarly, in Section 4.2, the plots in Figure 3 are only briefly mentioned without any discussion of the results.

2.3) Once again, in Section 5, the plot in Figure 4 (referred to as Fig. 5 in the text) is not truly discussed.

3) Proposed score:
3.1) As acknowledged by the authors themselves, Figure 4 suggests that the proposed score is not truly representative of the test accuracy. Additionally considering that, as mentioned before, the motivation behind this score was not very clear, I recommend the authors to remove this from their paper.

4) Related work: The related work section seems incomplete.
4.1) While Section 2.1 discusses some differentiable NAS methods, it ignores others even if they are used in the experiments, e.g., GDAS and DrNAS.

4.2) The papers listed in the general discussion of NAS methods fails to mention a number of works, such as Guo et al., Single Path One Shot, 2019; Cai et al., Once for All, ICLR 2020; Zela et al., Understanding and Robustifying Differentiable NAS, ICLR 2020; Luo et al., NAO, NeurIPS 2018; Zhang et al., ShuffleNet, CVPR 2018; Chu et al., FairNAS, 2019.

4.3) Other works that have proposed evaluation strategies for supernet-based methods would be worth discussing: Yang et al., NAS evaluation is frustratingly hard, ICLR 2020, Yu et al., An analysis of training heuristics in weight-sharing NAS, 2020.

4.4) Minor comment: In the discussion of some of the paper, as well as in the experiments, the references to the papers are often missing.

5) Case study: I am not entirely convinced by some of the arguments in the case study of Section 3.
5.1) The authors make the statement: "... without explanation of how or why these algorithms are able to positively interact." In fact, based on the results in Table 1 (which are not truly explained), these two algorithms do not positively interact; the best results are obtained by the method of Wang et al., 2021.

5.2) The statement "training with architecture weights does provide an improvement to the perturbation based selection method" is not entirely correct. It does so when the perturbation-based method is used with fixed $\alpha$, not if the perturbation method is used with the trained $\alpha$.

**Summary Of The Paper:**

This paper introduces an evaluation methodology for differentiable NAS methods. Specifically, it separates the search of these methods into two stages: Stage 1, which consists of training the supernet, and Stage 2, whose goal is to select an architecture. These two stages are evaluated independently, and the results of these evaluations are combined into a score whose goal is to provide information about how well any combination of one Stage 1 strategy with one Stage 2 strategy would perform.

**Summary Of The Review:**

Although this paper tackles an interesting problem, it is currently unconvincing. Many parts of the paper need to be clarified, the results need to be discussed, the proposed score does not serve the desired purpose, and the discussion of the related work should be improved.

---

> ### Author Response · Authors · 2021-11-23
> **Response to Reviewer XbXs**
>
> We thank the reviewer for helpful feedback and comments.  In response to the noted weaknesses we have:
> - We have performed major revisions to the text in an attempt to better communicate our motivation, perspective, and findings.  This includes simplifying our metrics and evaluation process.
> - We have modified the Stage-1 metric and figure to be more intuitive, and added an interpretation of the curve in the main text.
> - We have expanded our citations to broaden our literature review and make up for oversights in the previous manuscript
> - We have expanded and attempted to clarify the motivation and conclusions of our case study

---

### Official Review · Reviewer_eM7q · 2021-10-27

**Correctness:** 3
**Technical Novelty And Significance:** 2
**Empirical Novelty And Significance:** 3
**Recommendation:** 3
**Confidence:** 5

**Main Review:**

**Strong Points**
- It's a good point to study the integration of multiple DARTS variants, and see their joint influence.

**Weak Points and Questions**
- Missing reference for several methods like GDAS and DrNAS.
- For the writing part, the authors just put forward the table or statistics, but lack of proper explanation and justification.
I suggest that the authors add a conclusion part to discuss the analysis results for Section 4 and 5.
- Although such analysis is well-motivated, it is unclear to me how this conclusion can be extended to other spaces.
After all, 201 is a small space, and many NAS methods claim that they can find the optimal of the whole space.
I understand that other space is hard to analyze, but still the potential impact of the analysis might be limited.

**Summary Of The Paper:**

This paper doesn't propose a new method or algorithm.
Instead, they perform a detailed empirical study for the two components of differentiable NAS variants, i.e., supernet training and architecture selection.

**Summary Of The Review:**

This paper is a purely empirical evaluation paper, putting several advanced differentiable NAS methods together and study their joint effects.
However, the impact can be restricted as all studies are performed on a toy space. Additionally, I think the writing of this paper has a large room to improve, the authors should add some conclusions and discussions rather than just showing the tables.

---

> ### Author Response · Authors · 2021-11-23
> **Response to Reviewer eM7q**
>
> We thank the reviewer for helpful feedback and comments.  In response to the noted weaknesses we have:
>
> - We have included additional citations
> - We have added analysis to the experimental findings, both of our proposed metrics and of our case study, in hopes of better communicating our findings and perspective.
> - We have expanded the experiments to also include the ImageNet16-120 benchmark, as well as the larger DARTS-space search space on CIFAR-10.

---

### Official Review · Reviewer_JdoT · 2021-11-03

**Correctness:** 3
**Technical Novelty And Significance:** 1
**Empirical Novelty And Significance:** 2
**Recommendation:** 3
**Confidence:** 4

**Main Review:**

Overall, the studies are helpful, but it is unclear what are the key insights and innovations in this paper.

— Strengths —

1. Comprehensive description on the problems of supernet-based NAS;
2. Interesting perspective to divide NAS into two-stage process, and study the behavior for each of them;

—Weaknesses—

My main concern is the lack of insights: this paper is mostly focus on analyzing the two NAS stages, but it is unclear why those two metrics are important and what does it mean to future NAS search. Some of the results are obvious, such as Figure 3 (uniform tends to be slightly worse than others). It is unclear how we can better understand or further improve NAS based on the studies in this paper. Because of this, it is not obvious what is the core value of this paper.

I also have a few  other concerns:
1. Poor writing. Many concepts are explained well: for example, the abstract and introduction emphasize the novelty of eval metrics, but what are the metrics why they are novel?  To my knowledge, spearman correlation (used in Figure 2) has been widely used in NAS community.
2. Limited experiments:  as an analysis paper, it provides limited data on a simple nas-bench-201 benchmark. I would expect more experiments on more search spaces and algorithms.

**Summary Of The Paper:**

This paper studies the statistics of weight-sharing based neural architecture search (NAS) algorithms. In particular, it separates the training process into two stages: one is for supernet weight training and the other for architecture selection, and then it investigates the Spearman
correlation and other statistics. All experiments are based on NAS-Bench201.

**Summary Of The Review:**

1. Somewhat interesting studies for NAS by dividing the search into two-stages
2. Lack of insights;
3. Limited experiments.

---

> ### Author Response · Authors · 2021-11-23
> **Response to Reviewer JdoT**
>
> We thank the reviewer for helpful feedback and comments.  In response to the noted weaknesses we have:
>
>
> - We have attempted to better describe and frame our broader motivation and perspective on separate evaluation of performance estimation and architecture sampling in NAS.  We hope that additional detail on how this perspective affects our own interpretation of the findings in recent high-impact papers such as PC-DARTS may help to motivate the potential impact if the field at large were to adopt alternative evaluation metrics for NAS components (regardless of whether they are the ones we propose here -- or alternative that may be provided in the future).  We also have attempted to simplify and clarify the metrics for performance estimation and search strategies, both in the written interpretation and in the figures.
> -We have expanded the experiments to also include the ImageNet16-120 benchmark, as well as the larger DARTS-space search space on CIFAR-10.

---

### Official Review · Reviewer_SuQ3 · 2021-11-03

**Correctness:** 4
**Technical Novelty And Significance:** 3
**Empirical Novelty And Significance:** 3
**Recommendation:** 5
**Confidence:** 4

**Details Of Ethics Concerns:**

No ethical concerns.

**Main Review:**

## Strengths
- This paper is written pretty well - all of the decisions are carefully thought through and explained clearly in writing.
- The case study in Section 3 is interesting, and the ideas in Sections 4 and 5 are interesting and make sense.
- The topic of this paper, better methods for evaluating differentiable NAS, can have high impact in the NAS community, because there are many different differentiable techniques proposed.

## Points to be addressed
- One of my biggest concerns is that all experiments are only done on NAS-Bench-201 (and only on CIFAR10 and CIFAR100, not ImageNet16-120). NAS-Bench-201 is by far the smallest of the popular NAS benchmarks, with 15k architectures, and only 6k unique architectures. Therefore, there is no guarantee that the experimental conclusions will generalize. It would make the paper much better if experiments were done on another benchmark, such as NAS-Bench-1shot1 which contains 300k unique architectures, or NAS-Bench-ASR or TransNAS-Bench-101, etc.
- My other main concern is that the paper is a little bit too incremental for ICLR. While the ideas are nice, and I have not seen other papers talk about differentiable NAS in this way before, I am not sure it is quite enough content for ICLR. One thing that could help is if the ideas in this paper were used to find a conclusion that is even more novel, surprising, or insightful. (Or, use the new evaluations to propose a new algorithm other than Dirichlet-synflow?)
- Two of the validation methods, synflow and jacob_cov, have not been used as "Stage-2 Search" for differentiable NAS algorithms before, as far as I am aware? So are there only two types of stage-2 search from existing literature?
- The authors do not give a reproducibility statement or ethics statement. This was optional but encoraged by ICLR. I do not see ethical concerns, but having a reproducibility statment (and releasing code) would have made me feel more positive about the paper.
- Citing relevant work is sparse. The most glaring emission was not citing jacob_cov [1], even though this is used in the experiments ([1] came out before Abdelfattah et al. 2021). Also, not citing PC-DARTS [2] even though it is mentioned by name in Section 2.1. Other papers that seem relevant to talk about include (at least) [3], [4], [5].
- Finally (and least important), a few small comments about the figures:
  - It’s much better for viewing to have the figures be vectorized. E.g. save the plots as pdf's.
  - The x-axes in Figs 2 and 5 have to be cleaned up since there are overlaps.
  - In Figs 2 and 5, there are two different types of purples used. It is better to use more distinct colors.


[1] Mellor et al., Neural Architecture Search without Training, 2020.

[2] Xu et al., PC-DARTS: Partial Channel Connections for Memory-Efficient Architecture Search, 2019.

[3] Yu et al., How to Train Your Super-Net: An Analysis of Training Heuristics in Weight-Sharing NAS, 2020.

[4] Zela et al. Understanding and robustifying differentiable architecture search, 2020.

[5] Pourchot et al., To Share or Not To Share: A Comprehensive Appraisal of Weight-Sharing, 2020.

**Summary Of The Paper:**

The authors study the question of evaluating differentiable methods for neural architecture search (NAS). Differentiable techniques are popular in the NAS community, and many recent papers have given criticisms or improvements to various parts of the DARTS algorithm and related algorithms. The authors propose a new way of evaluating differentiable techniques, and new metrics, by separately evaluating architecture training and architecture selection, and evaluating the combinations of the two. The authors run experiments on NAS-Bench-201 and use their evaluation technique to identify the best combination of architecture training and selection.

**Summary Of The Review:**

I like the topic of this paper: devising better tools for evaluating differentiable NAS. And I have not seen the ideas in this paper presented in quite this way, which could be impactful in the community. However, there are unfortunately the weaknesses of only using NAS-Bench-201 and being a bit incremental. I recommend rejection.

---

> ### Author Response · Authors · 2021-11-23
> **Response to Reviewer SuQ3**
>
> We thank the reviewer for helpful feedback and comments.  In response to the noted weaknesses we have:
>
> - Expanded the experiments to also include the ImageNet16-120 benchmark, as well as the larger DARTS-space search space on CIFAR-10.
> - We have attempted to better describe and frame our broader motivation and perspective on separate evaluation of performance estimation and architecture sampling in NAS.  We hope that additional detail on how this perspective affects our own interpretation of the findings in recent high-impact papers such as PC-DARTS may help to motivate the potential impact if the field at large were to adopt alternative evaluation metrics for NAS components (regardless of whether they are the ones we propose here -- or alternative that may be provided in the future)
> - We have expanded our explanation of Stage-2 search to give a more complete account of the existing lack of variation in techniques and our use of Zero-cost NAS algorithms.
> - We have included a reproducibility statement and anonymously released our code.
> - We have included additional citations
> - We have attempted to better clarify figures

---

> > ### Comment · Reviewer_SuQ3 · 2021-11-29
> > **Thanks for the update**
> >
> > I thank the authors for the time they put into the rebuttal. In particular, the authors also added experiments for NAS-Bench-201 ImageNet (before there was only cifar10 and cifar100), and DARTS cifar10. I appreciate the authors releasing their code, especially with detailed README's. The authors also clarified the impact of their contributions and added a reproducibility section - note that it is allowed for this to go past the page limit of 9 pages, and can be at most one page, so more information can be added.
> >
> > I am raising my score from 3 to 5.

---

### Decision · Program_Chairs · 2022-01-20

**Decision:**

Reject

**Comment:**

All reviewers recommended rejection, and I agree.
I encourage the authors to follow the reviewers' recommendation and resubmit.